# APPLYING LANGUAGE MODELS TO ALGEBRAIC TOPOLOGY: GENERATING SIMPLICIAL CYCLES USING MULTI-LABELING IN WU'S FORMULA

## ABSTRACT

Computing homotopy groups of spheres has long been a fundamental objective in algebraic topology. Various theoretical and algorithmic approaches have been developed to tackle this problem. In this paper we take a step towards the goal of comprehending the group-theoretic structure of the generators of these homotopy groups by leveraging the power of machine learning. Specifically, in the simplicial group setting of Wu's formula, we reformulate the problem of generating simplicial cycles as a problem of sampling from the intersection of algorithmic datasets related to Dyck languages. We present and evaluate language modelling approaches that employ multi-label information for input sequences, along with the necessary group-theoretic toolkit and non-neural baselines.

## 1 INTRODUCTION

Mathematical progress often depends on the identification of patterns and the formulation of conjectures, which are statements believed to be true but yet to be proven. For centuries, mathematicians have relied on data, from early prime tables to modern computer-generated data, to aid in this process. While computational techniques have proven useful in some areas of pure mathematics (Appel & Haken, 1977), (Hales, 2005), the full potential of artificial intelligence (AI) in the field is yet to be fully explored.

In mathematics, AI has demonstrated remarkable abilities in tasks such as identifying counterexamples to conjectures (Wagner, 2021), accelerating calculations (Peifer et al., 2020), generating symbolic solutions (Lample & Charton, 2019), detecting structural patterns in mathematical objects (He, 2021), automated generation of formal proofs (Han et al., 2021), (Polu et al., 2022) and guiding human intuition to making hypothesis (Davies et al., 2021). These capabilities offer vast potential for expanding mathematical knowledge and advancing the field.

Inspired by these recent advanced in AI driven mathematical research, we propose a proof-of-concept framework for generating simplicial cycles in simplicial groups. This serves as an initial step towards a broader initiative of utilizing machine learning techniques to sample elements from the homotopy groups of spheres and general spaces. In the current work we apply our ideas in particular case of free simplicial group which models (loops over) a two-dimensional sphere. We adopt a simplicial and group-theoretic approach due to its combinatorial nature and computer-friendly representation.

The theoretical background for Wu's formula (4) is outlined in Section 3 with a more detailed exposition given in Appendix B. This formula expresses our object of interest as a certain quotient of the intersection of normal subgroups of free group. Generating elements from this intersection $\cap_i R_i$ (see notation (1) below) is the main topic of the present work. The key features of the problem include the similarity between $R_i$'s and sets of Dyck-$n$ words (allowing a suitable distribution on it to sample from) and a clear algorithmic procedure for determining if a given element $w$ lies within the intersection.

We develop the following non-neural-based baselines for this problem: random search, evolutionary method and greedy algorithm. These baselines are extensively described in Section 4.2. Concurrently, our main deep learning approaches described in Section 4.3 are based on language modelling and differ in the various usages of training dataset's meta information.

The goal of introducing the deep learning approaches is to increase the performance in terms of the number of generated words from the full intersection, while also achieving scalability of the model. This is crucial due to the rapid growth in computational complexity observed in the baseline algorithms, rendering them impractical for application in higher dimensions. The relevant set of metrics, datasets description and overall evaluation results are presented in Section 5 together with some relevant discussions pertaining to the findings.

## 2 RELATED WORK

The computation of homotopy groups of spheres stands as one of the most challenging problems in mathematics. Over the past century, the complexity of this problem has spurred the development of numerous sophisticated mathematical techniques. For instance, in the 1950s, Toda introduced the concept of secondary operations (now called Toda brackets), which allows him to compute the homotopy group of spheres up to a certain dimension (Toda, 1963). In the 1960s, unstable Adams spectral sequences were developed, providing a general framework for such computations and extending the range of results achieved by Toda (Adams, 1958). In the 1990s a fresh approach using Goodwillie tower of identity functor (Arone & Mahowald, 1999), (Behrens, 2012) was employed, leveraging modern techniques to recompute Toda's range.

In recent years, advancements have been made in computational and computer science approaches concerning the computation of homotopy groups.

A classical result by Brown (Brown, 1957) established that homotopy groups of spheres are finitely computable. However, it should be noted that the algorithm presented in Brown's work, as discussed in (Cadek et al., 2014), exhibits superexponential complexity. While this result lays a solid theoretical foundation for the development of algorithms and computational methods for computing these groups, the proof itself does not directly yield practical algorithms for their computation. Nonetheless, it has sparked new avenues of research and possibilities for researchers interested in devising effective computational methods for computing homotopy groups.

While the problem of computing even the weaker invariants like the rationalization $\pi_n X \otimes \mathbb{Q}$ for a general space $X$ is known to be $\#P$-hard (Anick, 1987), there have been notable advancements in algorithmic computation of homotopy groups and homology of spaces.

The authors of the Kenzo system (Sergeraert, 1994),(Rubio & Sergeraert, 2002), (Romero et al., 2006) since the 1990s have been actively developing an algorithmic framework based on spectral sequences to compute homotopy groups of certain classes of spaces. Their concept of *spaces with effective homology* was further enhanced in (Cadek et al., 2014), where a polynomial-time algorithm was introduced for computing $\pi_n X$ for a **fixed** value of $n$ within the framework of *simplicial sets with polynomial time homology*.

Besides the computational challenges, one limitation of current approaches is that $\pi_n X$ is typically computed as an abelian group, without providing a detailed account of its internal homotopical structure, such as a description of generators and relations in algebraic, geometric, or topological terms. However, there have been efforts (Filakovsky et al., 2018) proposing approaches to address this limitation.

The utilization of machine learning techniques in the calculation of homotopy groups of spaces is a relatively novel and emerging approach. So far, the corresponding research concerning both machine learning and algebraic topology, primarily dedicated to enhancing machine learning algorithms with ideas from algebraic topology, but lacks exploration of the inverse. The present work initiates the process of filling this gap.

## 3 GROUP THEORY OF WU'S FORMULA

To keep the preliminary exposition concise, we will provide a brief overview of the algebraic constructions employed in this paper, reserving more formal and detailed information for Appendix A. Wu's formula for $\pi_n S^2$, introduced in (Wu, 2001), combines the simplicity of group theoretic language of free group with a powerful combinatorial structure of simplicial groups. In our perspective, the group theoretic structures in Wu's formula, related to formal languages, presents a compelling target

for the application of language models. Additionally, the usage of simplicial groups opens up further avenues for incorporating machine learning to a more sophisticated mathematical constructions of simplicial homotopy theory, like the unstable Adams spectral sequence (Bousfield et al., 1966).

By *free group* $F$ on a set $x_1, \ldots x_n$ we will mean a set of words in letters $x_i$ (called *generators*) and their inverses with group operation of string concatenation. The identity $x_i x_i^{-1} = x_i^{-1} x_i = 1$ is assumed in $F$, here $1$ is an empty word. Additionally, we will denote by $x_0 = x_1 \ldots x_n$ the product of all generators $x_i$. We will further use the following notation: $u^v = v^{-1} u v$ called *conjugation* and $[u, v] = u^{-1} v^{-1} u v$ called *commutator* for $u, v \in F$.

We will be interested in generating words from the following subsets of $F$:

$$R_i = \{ x_i^{\pm y_1} \ldots x_i^{\pm y_k} \mid k \in \mathbb{N}, \ y_i \in F \}, \ i = 0..n. \tag{1}$$

The subgroups $R_i$, also denoted by $\langle x_i \rangle^F$, to emphasize the generating element, are called the *normal closures* of $x_i$'s. These subgroups can be conceptually understood as follows: for $i > 0$ the subgroup $R_i$ consists of words that become trivial after substituting $x_i$ for $1$, for example:

$$[x_1, x_2] x_1 \in R_1, \ [x_1, x_2] x_1 |_{x_1=1} = 1.$$

Similarly, $R_0$ consist of words which become trivial after any of the equivalent substitutions like $x_1 \mapsto x_n^{-1} x_{n-1}^{-1} \ldots x_2^{-1}$, which maps $x_0$ to $1$. Equivalently, $R_0$ consists of words which become trivial upon removal the cyclic permutations of $x_0$.

As a result, the intersection $\cap_{i=0}^n R_i$ consists precisely of words that become trivial when any of the (cyclic permutations of) $x_i$ is substituted with $1$. The objective of this paper is to propose a several deep learning generators of such words.

Since $[u, 1] = [1, v] = 1$ for all $u, v \in F$, it is clear that the elements of the form $[x_i, x_j]$ are in the intersection $R_i \cap R_j$ and, more generally, $[R_i, R_j] \subseteq R_i \cap R_j$. This idea can be iterated by introducing the *iterated commutators*

$$[u_1, \ldots, u_k] = [[[u_1, u_2], u_3] \ldots, u_k], \ u_i \in F, \tag{2}$$

and for a set of subgroups $\{ R_{i_1}, \ldots, R_{i_k} \}$ we have:

$$\prod_{\sigma \in \Sigma_k} [R_{\sigma(i_1)}, \ldots R_{\sigma(i_k)}] \subseteq \bigcap_{j=1}^k R_{i_j}, \tag{3}$$

where the left-hand side have a product over all permutations on $k$ letters, reflecting the symmetric nature of the intersection on the right-hand side. We will abbreviate the left-hand side of the inclusion (3) as $[R_0, \ldots, R_n]_S$ and call it a *symmetric commutator subgroup*.

The following surprising result is important for training our models:

**Theorem 3.1** (Wu (2001), Corollary 3.5). *For any* proper *subset* $\{ i_0, \ldots, i_k \} \subsetneq \{ 0, \ldots, n \}$ *the inclusion* (3) *is an equality.*

This implies that any partial (different from $\cap_{i=0}^n R_i$) intersection of $R_i$ can be easily described and, after defining the suitable distribution, sampled from. In (Wu, 2001) simplicial methods are used to demonstrated that the difference between the symmetric commutator subgroup and the **full** intersection, as expressed in formula (3), possesses a homotopical nature:

**Theorem 3.2** (Wu's formula, Theorem 1.7 in (Wu, 2001)). *For $n \geq 1$ there is an isomorphism*

$$\pi_{n+1}(S^2) \cong \frac{R_0 \cap R_1 \cap \cdots \cap R_n}{[R_0, R_1, \ldots, R_n]_S} \cong \frac{R_0 \cap [R_1, \ldots, R_n]_S}{[R_0, R_1, \ldots, R_n]_S}. \tag{4}$$

Although not computational in nature, the formula (4) gives an elegant combinatorial descrtiption of homotopy groups of $S^2$. This description motivates the present work.

*Remark* 3.3. It is important to note that the problem of classifying whether a given word $w \in \cap_i R_i$ is *trivial* (is an element of the denominator of formula (4)) or *non-trivial* (contributes to a non-zero element of $\pi_* S^2$) is almost as hard as the problem of computing homotopy groups themselves. To

Table 1: Elements from $\cap_{i=0}^{n} R_i$, that are not contained within $[R_0, \ldots, R_n]_S$

| $n$ | $\pi_{n+1}(S^2)$ | Non-trivial element |
|---|---|---|
| 2 | $\mathbb{Z}$ | $[x_1, x_2]$ |
| 3 | $\mathbb{Z}/2$ | $[[x_1, x_2], [x_1, x_2 x_3]]$ |
| 4 | $\mathbb{Z}/2$ | $[[[x_1, x_2], [x_1, x_2 x_3]], [[x_1, x_2], [x_1, x_2 x_3 x_4]]]$ |

our knowledge, there are no known algorithms or approaches to tackle this problem in a general setting, each elements requires an individual method for classification. For instance, for $n = 3, 4$ the group theoretic form of the corresponding generators of the homotopy groups is known (Mikhailov, 2021): In our research, we aim to generate elements from the numerator of Wu's formula without providing any information about its denominator to the model, thereby enforcing it not to "learn" the pattern of the trivial words in the denominator. Furthermore, during the training phase of our algorithms, we do not intend to utilize the knowledge of non-trivial elements listed in Table 1, since we want our approaches to be scalable. However, we can consider utilizing the non-trivial elements and the denominator in the evaluation phase to assess the performance of our approaches.

# 4 GENERATING ELEMENTS FROM THE INTERSECTION OF NORMAL CLOSURES

## 4.1 CREATING DATASETS FROM NORMAL CLOSURES AND THEIR COMMUTATOR SUBGROUPS

Generating a consistent synthetic dataset is of paramount importance for data-driven projects. In this regard, one of the biggest challenges is to ensure that the dataset is not only large enough but also exhibits a reliable structure. In our project, we encountered a similar challenge while constructing a training dataset comprising words from $R_i$ and their partial intersections.

In order to effectively train our models, we need to have a certain amount of control on properties of training dataset, for example being able to control to some extent the distribution of lengths of the words and their internal structure.

When it comes to generating words from $R_i$, we discovered that using a *naive* sampling approach, where each $y_k$ in formula (1) is sampled independently, yielded less desirable properties in the generated words: the adjacent letters of different conjugators $y_k$, $y_{k+1}$ do not interact with each other, resulting in reduced variability within the generated dataset.

To address this issue, we developed an alternative description of $R_i$, based on balanced bracket sequences and Dyck-$n$ languages. Specifically, to generate a word from $R_i$, we can uniformly sample a balanced bracket sequence (Atkinson & Sack, 1992) and then replace every matching pairs of brackets with words $a$ and $b$ such that the word $ab$ is either identity or the rotation of the normal closure's generator. For instance, if $n = 3$, then in case of $R_0$ the opening bracket $a$ could be $x_2$ and the corresponding closing bracket will be $b = x_3 x_1$. We refer to this sampling method as *bracket-style sampling*.

We use the notion of Dyck paths to analyse the difference between naive and bracket-based sampling. By *Dyck-$n$ word* we will mean a balanced bracket sequence with brackets of $n$ types. For Dyck-$n$ word $b$ the corresponding *Dyck path* $p(b)$ is a plot that depicts the variation in the number of unbalanced brackets as a function of the position in the word. By *valley* on Dyck path $p(b)$ we will mean a local minimum which is greater than zero.

Figure 1 illustrates the difference in distribution of valleys for naive and bracket-style sampling methods. While the distributions of word lengths remain similar, the bracket-style sampling method shows more variety in possible number of valleys in corresponding Dyck paths. Note that each valley corresponds to a mutual cancellation of letters in adjacent conjugators $y_k$ and $y_{k+1}$.

Now, to generate elements from iterative commutator subgroups like $[R_{i_0}, \ldots, R_{i_k}]$ we use the following procedure. First, we sample a collection of random binary trees that represent products of various commutators. Then for each leaf $j$ of every tree the element of $R_{\sigma(j)}$ is sampled using bracket-style sampling, where $\sigma \in \Sigma_k$ is a random permutation. Note that although commutators of arbitrary complexity can be expressed as product of iterated commutators of the form (2) using

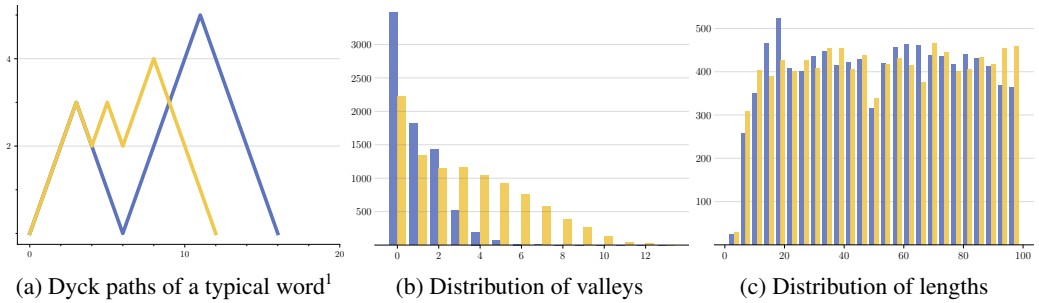

(a) Dyck paths of a typical word[1]  (b) Distribution of valleys  (c) Distribution of lengths

Figure 1: Comparison of naive (blue) and bracket-style (yellow) sampling. Sample of $10^4$ elements of $R_1$ with $n = 3$ with a maximum length of 100.

Hall-Witt identity (see Appendix A for details), we found that sampling not only iterated, but arbitrary commutators, such as $[[x_{i_1}, x_{i_2}], [x_{i_3}, x_{i_4}]]$ yields better results.

Finally, to ensure that the length distribution of words is equal among various partial intersections, we employed the Kolmogorov-Smirnov test. Further details regarding supplementary properties of the sampling methods can be found in Appendix D.

### 4.2 BASELINES

We choose a range of non-neural based algorithms to approach a problem of generating elements from the intersection $\cap_{i=0}^{n} R_i$, given algorithms for generating elements from each individual $R_i$.

**Random search**  Guided by Theorem 3.1, the most straightforward and readily implementable method is to randomly sample words belonging to $[R_0, \ldots, \hat{R}_i, \ldots, R_n]_S$ and check whenever such these words lie in $R_i$. Notation $\hat{R}_i$ here means that $R_i$ is excluded from the symmetric commutator subgroup. We are performing this procedure for all $i \neq 0$ simultaneously. Already in random search the difference between naive sampling and bracket-style sampling from $R_i$, discussed in Section 4.1, can be seen in terms of number of generated words from intersection per batch (see Appendix D for additional experiments).

**Evolutionary method**  As described in Section 3, the words in $\cap_i R_i$ have are characterized by the following property: replacing any of $x_i$ with 1 will reduce the word to 1. This criterion underlies the extremal reformulation of the task of generating words from the intersection of $R_i$ and the following algorithm.

For a normal closure $R = \langle v \rangle^F$ of a word $v$ and a word $w$, let the *algebraic distance* $d(w, R)$ be defined by the length of word after throwing away all occurrences of cyclic permutations of $v$ from $w$. Note that $d(w, R) = 0$ iff $w \in F$. This definition resembles algebraic distance (Taubin, 1991) between a point $x$ and the kernel of a map $f : \mathbb{R}^d \to \mathbb{R}$, defined by $d(x, \ker f) = |f(x)|$.

Next, let's introduce the non-negative objective function

$$d(w) = \sum_{i=0}^{n} d(w, R_i)$$

and consider the corresponding discrete optimisation problem $\min_{w \in F} d(w)$, solution to which will give an element from $\cap_{i=0}^{n} R_i$.

To solve this discrete optimisation problem we apply a modification of the $(1 + 1)$-evolutionary algorithm (Droste et al., 2002) with mutation replacing each of randomly chosen letters in the word with (uniformly) random sampled generator or its inverse. It is worth mentioning that one of our metrics (see Section 5) is inspired by the notion of distance between the intersection $\cap_i R_i$ and a given word.

---

[1]The words are: $y^2[x, z^{-1}]y^{-2}[z^{-1}, p][x, z^{-1}][p, z^{-1}]$ with 0 valleys and $p^2[z, x][pxp, y]p^{-2}$ with 2 valleys

**Greedy algorithm**   Experiments with language models (Section 4.3 below) that generate a full word based on a prefix inspired us to find a non-neural based iterative procedure of appending a next token to a given prefix to generate a full word from the intersection. For each normal closure $R_i$, a stack $S_i$ is created from the given prefix. Specifically, $S_i$ is obtained by removing $x_i$, its inverse, and cyclic permutations (in the case where $i = 0$) from the prefix.

To determine the next token in the sequence, the algorithm employs a greedy strategy based on the current tops of the stacks. Each token is assigned points based on two mutually exclusive criteria. The first criterion is that the token candidate should decrease the length of a particular stack. For instance, if the top of a stack $S_i$ is $x_j$, then the next token candidate $x_j^{-1}$ would receive points. The second criterion is that the token candidate should not increase the length of any stack. For example, the token $x_i$ does not increase the length of $S_i$ since it is already excluded from it. Tokens that would reduce the length of the given prefix itself are excluded and assigned zero points. The algorithm then selects the next token with the highest score.

For illustrative purposes we included a toy example of application of this method in Appendix D

## 4.3   DEEP MODELS

**General architecture details**   We now move to the neural-based algorithms for sampling elements from $\cap_i R_i$. We mostly use decoder-only transformers (Radford & Narasimhan, 2018), (Vaswani et al., 2017) as a main architecture for our models. We will briefly review its features relevant to our problem at hand.

Decoder-only models are often used in tasks where the input sequence is fixed or pre-processed, such as language modelling (Jurafsky & Martin, 2000), text generation (Li et al., 2022), which aligns with our task of generating words as sequences of tokens that matches a certain target distribution.

We adopt the classical auto-regressive approach, where each token in the output sequence is generated based on the previous tokens and the decoder's internal state. Specifically, at each time step, the decoder takes in the previous token in the output sequence, along with a context vector that summarizes the encoded input sequence, and generates a new token using a softmax (M., 2006) function.

We also utilize standard modifications of decoder-only transformer models, such as causal masking, which ensures that the model only attends to previous tokens in the output sequence, and positional embeddings, which provide information about the position of each token in the sequence. These modifications help the model generate high-quality and coherent output sequences (Brown et al., 2020).

To train such a model we use a cross-entropy loss (Bengio, 2008):

$$L(y, \hat{y}) = - \sum_k \sum_i y_{k,i} \log(\hat{y}_{k,i}) \tag{5}$$

where $y_{k,i}$, is a true probability distribution (of $k$-th token in the sequence having value $i$), $\hat{y}_{k,i} = \mathbf{Pr}_\theta(s_k = i \,|\, s_1 \dots s_{k-1})$ is the corresponding predicted probability distribution (which depends on model parameters $\theta$) , $s_1, \dots, s_t$ is a sequence of tokens from the training dataset.

We use generators $x_k$ and their inverses $x_k^{-1}$ as tokens, together with the special tokens: $\langle\mathbf{bos}\rangle$ to identify the beginning of the sequence, $\langle\mathbf{eos}\rangle$ to identify the end of the sequence, and $\langle\mathbf{pad}\rangle$ to identify *empty* tokens after the end of the sequence (ignored in formula (5)). The inclusion of the $\langle\mathbf{bos}\rangle$ token has been shown to significantly improve the performance of attention-based models in Dyck-$n$ language recognition (Ebrahimi et al., 2020), see also (Hahn, 2020), (Weiss et al., 2021).

Finally, to sample from models we use two configurations: beam search (Tillmann & Ney, 2003) with the repetition penalty (Keskar et al., 2019) and nucleus sampling (Holtzman et al., 2020).

### 4.3.1   METHODS

The key ingredient in our deep learning-based generating approaches is multi-label for each word in the training dataset. Multi-label of the word $w$ provides information about the membership of $w$ in each of the subgroups $R_i$:

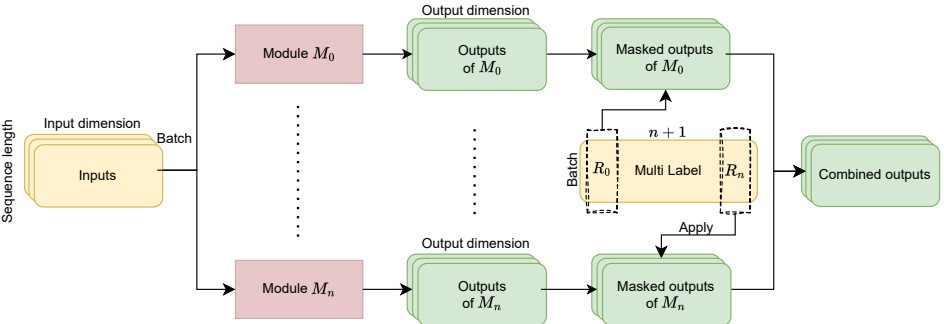

Figure 2: Architecture for a generalized masking model. If each module $M_i$ represents an attention head (or a group of attention heads), then we get a masking model. If the modules $M_i$ represent the complete transformers, we obtain an ensemble model. In both cases the combined outputs is a result of application of the linear transformation (trainable or constant) to the stacked masked outputs.

**Definition 4.1.** For an element $w$ of a free group on $n$ generators, the *multi-label* is a binary array of length $(n + 1)$ such that its $i$-th position decodes whether $w$ contained in $R_i$ or not.

We developed a range of approaches which are different from each other in a way how they treat multi-labels.

**Ignore**    This procedure involves the deliberate exclusion of multi-label information during both the training and inference phases. This approach is based on the underlying assumption that the model exhibits sufficient capacity to effectively generalize from a dataset containing words associated with various multi-labels. In essence, this can be described as a zero-shot generation, a problem that has been extensively studied in recent years (Wang et al., 2019), (Cao et al., 2020).

**Mask**    The second approach involves selectively masking specific components of the model in accordance with the corresponding multi-labels. In this method each attention head is assigned a normal closure, and during the training phase we mask the attention heads that do not correspond to normal closures containing the given word by multiplying their output with zero matrix. In the inference phase we do not mask anything to prompt the model to generate from the complete intersection. Using this masking approach we try to help the model to distribute the knowledge about the normal closures across its parts.

*Remark* 4.2. It is worth noting the *ensemble technique*, which is a particular case of the masking approach. This technique involves training of $n + 1$ models on each normal closure. During the inference phase, the given prefix is fed to each model, and next token is sampled from the joint distribution, which is essentially the sum of the outputs distributions

$$f(w) = \text{Softmax} \left( \sum_{i=0}^{n} M_i(w) \times \chi_{\{w \in R_i\}} \right),$$

with $\chi$ being a characteristic function of a subset. This approach is functionally equivalent to creating a model that comprises $n + 1$ transformers, with the logits of the last output layers being summed together, as Figure 2 illustrates. During training, the outputs of each model in ensemble are correspondingly masked according to the multi-label.

**Prompt**    Our third approach utilizes multi-label information as a *prompt* for the model. To facilitate this, we expand the vocabulary by introducing tokens for each normal closure $R_0, \ldots, R_n$ along with a token $\langle : \rangle$ that serves as an identifier for the end of the prompt. Prior to inputting the words to the model, we prefix the corresponding prompt to every word. For instance, in the case of a word $x_1 x_2 \ldots x_t$ with multi-label `[1, 1, 0, 0]`, we provide the model $R_0 R_1 : x_1 x_2 \ldots x_t$ as input. In the evaluation phase we prompt the model with $R_0 \ldots R_n$ :, indicating our intention to obtain a word from the complete intersection. Prompting has become increasingly popular in recent years (Liu et al.,

2021) as a means of fine-tuning pre-trained models for specific tasks such as question-answering, summarization, and sentiment analysis. By providing targeted prompts, the model's output can be optimized for specific use cases, making it more accurate and efficient at performing specific tasks.

**Negative Baseline**    We can collect the dataset of words from the $[R_0, \dots, R_n]_S$, i.e. a denominator of Wu's formula (4) and train a model to generate elements from it. Although we can not use this method in our comparison, since it was trained only on trivial words (see terminology of Remark 3.3), but we can employ its metrics in order to gain new insights about our primary methods. Furthermore, negative baseline can be used for so-called *negative knowledge distillation* (Gou et al., 2021): fine-tuning one of the primary models to predict the outputs that are **not** predicted by the negative baseline.

## 5    EVALUATION

Before discussing the results of the evaluation of the proposed models, we explain the relevant metrics and datasets, with other technical implementations details given in Appendix C.

**Metrics**    There are families of metrics commonly used in natural language processing, which, following (Celikyilmaz et al., 2020), may be divided in three groups: human-centric evaluation metric, automatic metrics and machine-learned metrics. We decide to concentrate our attention on automatic metrics, since our primary objective is to understand the quantitative performance of neural-based algorithms in a task of generating the words from $\cap_i R_i$.

Our main metric, the *completion ratio*, is an average number of the successfully generated words per batch. Specifically, for a batch of randomly generated words prefixes, the model generates a set of possible outputs based on each prefix. The number of generated outputs that simultaneously belong to all normal closures $R_i$ is then computed, and this number is divided by the size of the batch. A higher completion ratio signifies a greater proficiency of the model in generating desired outputs.

In addition to CE loss (5), we also utilize a *reduction ratio* for guiding the training process. This metric measures the distance (see Section 4.2) from the generated output to the given $R_i$ divided by the length of the input, batch averaged and averaged over all $i$.

Reduction ratio sits "in between" the loss and the final completion ratio metric: once the loss on the validation dataset has stabilized, and the completion ratio remains relatively low, the changing rate of the reduction ratio becomes informative. This changing rate indicates how the model's output gradually approaches the numerator of Wu's formula (see Figure 3).

*Remark* 5.1.  We also notice that although for future projects of generation *non-trivial* words human evaluation is still out of scope due to our constrained knowledge of the intricate structure of the homotopy groups. While there is a relative limited number of mathematicians deeply immersed in this topic, the machine-learned metrics (for instance, probability of the word being contained in the symmetric commutator subgroup) may be beneficial and play significant role.

**Datasets**    In order to maximize the performance of our models, specifically in terms of the number of generated words from random prefixes, we adopt an approach where overfitting is not a primary concern. To achieve this, we employ an infinite, online generated training dataset that encompasses words from $[R_0, \dots, \hat{R}_i, \dots, R_n]_S$ for all $i$, i.e. the multi-labels associated with these words contain only a single zero. The validation dataset is also generated in an online fashion, drawing samples from the same distribution. For a negative baseline model, described in Section 4.3.1, both training and validation dataset consist of words from the full symmetric commutator subgroup $[R_0, \dots, R_n]_S$.

For an auxilary tests, we have assembled small datasets consisting of cyclic permutations of generators of $\pi_n S^2$ for $n = 3, 4$ from Table 1. By evaluating our metrics on these cyclic permutations, we can assess the models proficiency in generating *non-trivial* words.

### 5.1    RESULTS

We compare our deep learning approaches with non-neural-based baselines using completion ratio, see Table 2. We also report the metric values on validation during training progress (Figure 3), to

Table 2: Completion ratio of developed algorithms for various number of generators $n$.

|  | Random | Evo | Greedy | Ignore | Mask | Prompt |
|---|---|---|---|---|---|---|
| $n = 3$ | 0.058 | 0.035 | 0.479 | **0.36** | 0.16 | 0.30 |
| $n = 4$ | 0.032 | 0 | 0.016 | 0.356 | 0.584 | **0.684** |
| $n = 5$ | 0.025 | 0 | 0.004 | 0.263 | 0.487 | **0.550** |

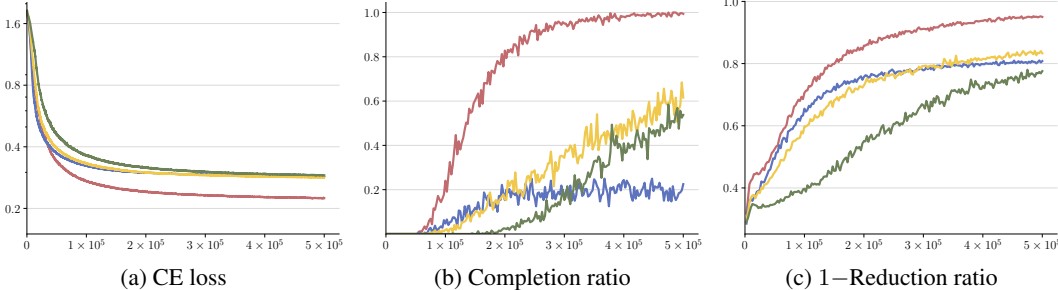

   (a) CE loss               (b) Completion ratio            (c) $1-$Reduction ratio

Figure 3: Illustration of the learning process for *negative baseline* (red), *prompt* (yellow), *masking* (green) and *ignore* (blue) methods, $n = 4$.

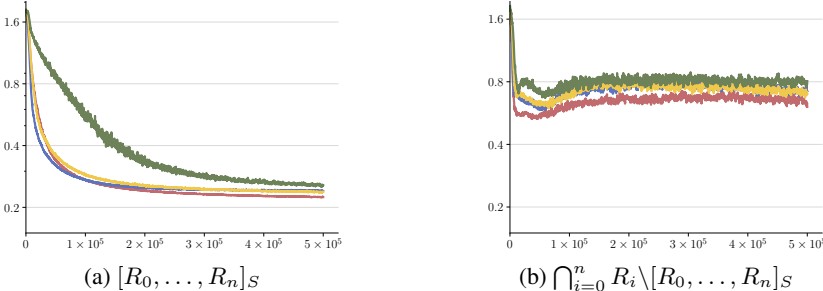

         (a) $[R_0, \ldots, R_n]_S$                  (b) $\bigcap_{i=0}^{n} R_i \backslash [R_0, \ldots, R_n]_S$

Figure 4: Comparison of CE loss per epoch on validation for the trivial and non-trivial words, $n = 4$.

illustrate the difference between various multi-label approaches. We report our results for $n \leq 5$ generators, due to the available computation resources. We also notice (see discussion below), that for a higher number of generators the high length of words from the full intersection $\cap_i R_i$ suggests to use another, more sophisticated representation of words.

Table 2 provides compelling evidence of the scalability and superior performance achieved by language models when compared to the baselines. These results signify a promising avenue for future applications of such models in understanding simplicial homotopy groups. In terms of comparing the performance of the methods between each other, the prompt-based approach outperformed the other methods on the validation dataset. Finally, Figure 4 illustrates that the non-trivial words are significantly more challenging (as expected) for the models to generate.

## 6 CONCLUSION

We evaluate perspectives of using machine lerning in generating cycles in simplicial groups. From a machine learning perspective, the problem can be reformulated as the task of generating elements from the intersection of subsets of (a form of) Dyck-$n$ words. Some classical algorithms are implemented, some are developed from scrath and used as baselines for deep learning algorithms. Built of the idea of ensemble of generators, the additional multi-label information is added to the training dataset, which allows the single model to work as a generalization of an ensemble. Resulting models, in contrast to baselines, are scalable, and will serve as a building blocks for future algorithms specialized in sampling from homotopy groups of spaces.

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
