# OpenReview forum: "Applying language models to algebraic topology: generating simplicial cycles using multi-labeling in Wu's formula"
_ICLR.cc/2024/Conference — Submitted to ICLR 2024_

### Official Review · Reviewer_VtUd · 2023-10-29

**Soundness:** 3 good
**Presentation:** 2 fair
**Contribution:** 2 fair
**Rating:** 3
**Confidence:** 3

**Summary:**

The paper considers using machine learning approaches to study a problem in algebraic topology, namely the problem of sampling elements from the numerator of Wu's formula. It does so by proposing several variants of natural language models to generate possible words which may be elements of this numerator set, after generating a synthetic dataset based on sampling from Dyck paths.

**Strengths:**

The paper has an original premise in using tools from machine learning to study a specific problem in algebraic topology which requires generative modeling. This may be of  interest to NLP researchers who wish to understand the breadth of applications which are suitable for their architectures. Table 2 also clearly shows that the proposed methods outperform the baselines.

**Weaknesses:**

My main concern with this paper is its relevance within the machine learning field.

From a presentation perspective, the work emphasizes the mathematical contributions by framing the problem with Wu's formula, rather than focusing on the machine learning methodology. Because the setup of Wu's formula is quite complicated for anyone unfamiliar with algebraic topology, the paper may be difficult to parse for all but a few attendees of the conference. As written, it's hard to understand some of the notation (e.g., $[x_i, x_j]$ is defined for elements of $F$, but $[R_i, R_j]$ is not for the subgroups, which is a core component of the formulas). Further, because the paper spends so much time on these preliminaries, it is difficult to understand which machine learning methods are used and why. For example, the training procedure is difficult to understand.

From a more substantive perspective, the paper doesn't provide convincing evidence that this use of machine learning can provide meaningful progress for the problem in the domain of mathematics. For example, scalability of the method is highlighted, but the experiments only go up to $n = 5$ due to computational constraints (p. 9). It's not immediately clear from the writing that this contribution can be built upon for further research.

Ultimately, the paper may be of more interest to a pure mathematics community.

Some secondary concerns:

1) The use of the completion/reduction ratio metrics could be better justified: can this distinguish whether the model is suggesting a wide variety of elements from that set or simply repeating the same element multiple times?

2) The notation throughout the work is often difficult to understand: for example, the $\pm$ notation in (1) is unclear, and it's not immediately clear how the $y_{k,i}$ terms in (5) are related to the notation $y_i$ often used to describe elements of the set $F$

**Questions:**

1) Related to secondary concern 1 above: how do we know that the model is generalizing well, i.e. finding words which are not from the training set?

2) Are there simpler generative models which could work well for this task?

---

> ### Author Response · Authors · 2023-11-20
>
> **relevance within the machine learning field**
>
> We hope that our work will be of interest for people working not only in applications of machine learning in math, but to a machine learning researchers who are interested in conditional generation and generation of synthetic and algorithmic languages. A machine learning researcher may found interest in a fact that in our work the target distribution, namely the full intersection, is not directly accessible to us and is not employed in the training process in any way. The only exception is its symmetric commutator part, which is utilized exclusively in training the negative baseline model. We also compare various modes of ensembling multiple generators which we believe is not domain specific.
>
>
> **difficult to understand which machine learning methods are used and why**
>
> Sorry for this. We attacked a problem first by constructing a series of deterministic and optimization-based approaches. When they became insufficient we turned our attention to deep learning models. We can definitely expand on a motivation for using attention based models for the camera-ready version (some of this motivation is related to a research on generating Dyck languages mentioned in Section 4.3). In short, we used a range of simpler neural network architectures first (described in Appendix D.1), and then moved to the attention-based models. Starting from ensembles we moved to prompting techniques that showed the best results.
>
>
> **the training procedure is difficult to understand**
>
> Thank you for pointing it out! You are right, we should wrote it more carefully. The training procedure fully conicides with the standard training algoirthm for autoregressive language models employing cross-entropy as a loss function.
>
>
> **meaningful progress for the problem in the domain of mathematics? scalability?**
>
> Our main goal was to establish a new machine learning framework for attacking the problem of computing homotopy groups of spaces. In this context, the domain of machine learning offers a unique arsenal of tools, particulary flexible generative models. We are sorry that from the paper it is unclear that research can not be expanded, we think the opposite. Due to size constraints we exclude the "future research" section from this version of the paper, we will definitely include it in the final version. For example, now we are working on different tokenization (on the level of commutators, not the generators of free group), which will help with computational constrains. Our main direction for the research is to use the following regularization technique: we increase the loss of the model's prediction if it is very similar to the predictions of the 'negative baseline model'.
>
>
> **completion/reduction ratio can distinguish ... repeating the same element multiple times?**
>
>
> During the test time we sample a batch of distinct prefixes of certain length. Then for each prefix we say that the model "completed" this prefix if it was able to generate a word from the full intersection using this prefix as a prompt. Our metrics aggreate the result across these distinct prefixes. Taking this description into account, it is easy to construct a degenerate example with a high score. Having a prefix $p$ one can continue it with $wp^{-1}$, where $w \in [R_0\dots R_n]_S$ is a fixed word. Such model would score perfectly. We are now working on the additional metrics that would help us to detect such behaviour. We think of employing pairwise similarity coefficients across the batch sample.
>
>
> **how do we know that the model is generalizing well, i.e. finding words which are not from the training set?**
>
> We agree that the question of generalization is very important. We guarantee that the training dataset does not contain words from the *full* intersection and as you can see our model is capable of generating elements from the *full* intersection. So, we believe that our model is able to generalize well. However we might need to investigate this question further. One way of doing so is the following. We can sample $m$ prefixes of length $l$ before training and put them away as the test dataset. During the training we would sample words that do not start with these test prefixes. Using this strategy we can further evaluate the generalization performance of our model.
>
>
> **Are there simpler generative models which could work well for this task?**
>
> As mentioned in Appendix D.1, LSTM ensemble works relatively well, but is still outpermormed by our final model.

---

> > ### Comment · Reviewer_VtUd · 2023-12-05
> > **Appreciate the responses and additional work, but I don't feel the paper is ready for publication**
> >
> > Thank you so much for the detailed response to my concerns. The additional plot provided in the supplementary material is useful and helps to clarify some of the technical concerns I had about the work.
> >
> > That being said, I agree with Reviewer LHhA's assessment that the work needs to be expanded in order to be appropriate for ICLR. Along with building towards a more practical solution towards the problem at hand, I believe the presentation of the work needs to be improved for a machine learning audience. Namely, the technical presentation is seated in algebraic topology which makes it difficult for the vast majority of machine learning experts to understand the learning problem at hand. As a result, I don't think the paper provides a good enough foundation for interdisciplinary discourse between algebraic topologists and machine learning scholars.
> >
> > The premise of this work is innovative and I believe that with significant improvements (both technically and presentation-wise), it can be an important interdisciplinary contribution to the literature. However, at this point I don't feel that the paper is ready for publication.

---

### Official Review · Reviewer_LHhA · 2023-11-01

**Soundness:** 3 good
**Presentation:** 2 fair
**Contribution:** 3 good
**Rating:** 3
**Confidence:** 4

**Summary:**

In this work the authors propose the use of language modeling techniques and architectures as a means for generating elements of the homotopy group of the sphere. First, the authors cite Wu's formula, which provides a connection between the homotopy groups of a sphere and a quotient group of particular combinations of free groups. Specifically, the $n+1$ homotopy group is given by a quotient group, the numerator of which is an intersection of subsets $R_i$, called *normal closures*, whose elements are strings comprised of characters $\\{x_i^{\pm 1}\\}_{i=1}^n$.

The authors claim that generating words in the intersection $w\in \cap_{i=0}^n R_i$ is not straightforward, however any partial intersection (i.e. omitting at least one $i \in \\{0,\ldots, n\\}$) has an explicit description and can be sampled from. The authors propose a particular sampling method, and propose to use it to generate words with multi-labels as to which $R_i$ they belong. They then train a decoder-only transformer on these sampled words, leveraging the multi-labels via attention masking or prompting (eg. if $x_1 x_2^{-1} \ldots x_t \in R_0 \cap R_1$, they train the model on $R_0 R_1: x_1 x_2^{-1} \ldots x_t$). They then exploit these mechanisms at inference time to generate elements of the full intersection.

The authors propose a number of symbolic baselines (random search, an evolutionary method, and a greedy algorithm) which attempt to generate elements in the full intersection. They compare these methods to their approaches using language modeling and measure the *completion ratio*, which is essentially the percentage of generated words which are actually in the full intersection. In all but one case they find the language model is far better at generating words in the full intersection than any of the synthetic approaches.

**Strengths:**

The paper's greatest strength is in it's originality. To be fair, this is not a line of work I am intimately familiar with, however a literature review on the topic suggests to me that this work is novel in both application and method.

The authors are also operating in a highly technical application area (homotopy theory), and to their credit I feel they do an admirable job conveying the necessary information without getting bogged down in details, and include additional background material in the appendix.

**Weaknesses:**

The largest weakness, in my opinion, is that the significance of this work is unclear. The authors motivate the work by a connection with homotopy theory, however their approach does not generate elements of the homotopy group itself but rather just the numerator (i.e. full intersection of normal closures). Moreover, there is no notion of the *coverage* of this generative procedure. More specifically, based on the coverage metric, a constant function which simply always returned a fixed element $w \in \cap_{i=0}^n R_i$ would score perfectly, but this would not be useful in any way.

The lack of any notion of "coverage" means that, even if their approach *did* generate elements of the homotopy group with some high probability, it is unclear to me exactly how those words could be used to gain any insight into the structure of the homotopy group itself, because we would not have any assurance that it was covering the full set of potential words in the homotopy group.

The paper also needs significant editing before publication. There are many mistakes and typos, some of which lead to unparsable text.

**Questions:**

1. Can you explain more what you meant by "thereby enforcing it not to 'learn' the pattern of the trivial words in the denominator" at the end of Section 3? I don't understand this statement, because (as I understand it) you were not doing anything in particular to encourage it not to learn the trivial words, and indeed the results in Figure 4 suggests that the models very much did predominantly learn trivial words.
2. In your discussion on random search (in section 4.2) it was mentioned that the difference between naive sampling and bracket-style sampling from $R_i$ was evident in the number of generated words from intersection per batch. Could you please expand on this - what does the number of generated words from intersection per batch of random search imply about naive vs. bracket-style sampling?
3. Could you provide a simple explanation of why the bracket-style sampling is preferred to naive? (a) It is stated in section 4.1 that "adjacent letters of different conjugators $y_k, y_{k+1}$ do not interact with each other, resulting in reduced variability within the generated dataset" but this notion of "variability" has not been defined. Also, isn't this straightforward to solve by introducing dependencies between $y_k$ and $y_{k+1}$? (b) I understand the statistics calculated in Figure 1, but I don't understand why the distribution for the bracket-style sampling is "better" than naive. Also, if one wanted a more uniform distribution of valleys, couldn't we just always include the inverse of every word?
4. Could you provide quantitative evidence that the LLM results are "covering" the space well? I realize this might be challenging, the metrics which come to mind often involve knowing the ground-truth $cap_{i=0}^n R_i$ in order to provide a percentage, but without this there's seemingly no way to know if the model is not simply exploiting some degenerate pattern to solve the task easily but in a useless way. Would it be possible to provide such a metric if we limit ourselves to words with length less than some threshold?
5. You mention in the "Datasets" section that the training dataset is infinite and generated online, and that the validation dataset is also generated in an online fashion. Does this mean there is potential for train/test overlap?

**Typos / Minor Suggestions** (small selection)
* $y_i$ in equation (1) should probably be $y_j$ (I assume the subscript here has no relation to the subscript of $R_i$).
* The set $[R_i, R_j]$ is not defined. I assume $[R_i, R_j] = \\{[u,v] \mid u \in R_i, v \in R_j\\}$.
* I wasn't able to parse the first sentence of Remark 5.1.

---

> ### Author Response · Authors · 2023-11-20
>
> **the significance of this work is unclear**
>
> Our work aims to transition the problem of computing homotopy groups (with no general solution currently) into a domain of machine learning where a broader array of methods can be applied. In our opinion setting such framework is important.
>
> **a constant function ... would score perfectly**
>
> You are right. In the answer to VtUd we described a construction of the degenerate example very close to a constant function that fools the metric. We are working on incorporating word similarity-based metrics to exlude such degenerate cases.
>
> **not doing anything in particular to encourage it not to learn the trivial words**
>
> By "enforcing it not to 'learn' the pattern of the trivial words" we meant that we ensure that no words from $[R_0\dots R_n]_S$ are in the training dataset. It is done by manually checking that no words in the training dataset are from the full intersection as well.
>
> **Figure 4 suggests that the models very much did predominantly learn trivial words**
>
> Non-trivial words (on which the loss in computed in Figure 4) are obtained by cyclic permutations of generators for the known non-trivial element (Table 1 of the paper). By multiplying this word by some other word from $[R_0\dots R_n]_S$ we can obtain another representative with much lower loss. Our aim here was to highlight a highly non-typical structure of the distiguished representative found by hand.
>
> **naive sampling and bracket-style ... the number of generated words**
>
> Please refer to Figure 1 in the new supplementary, which illustrate the benefits of using bracket-style sampling in terms of greater variability. It can be seen that for bigger batches bracket-style sampling provide better coverage of $R_i$.
>
> **why the bracket-style sampling is preferred to naive?**
>
> Because the samples are more diverse. A side comment: your inquiry into the differences between these methods is greatly appreciated, as it prompted further investigation on our part. We realized that we can rewrite (non-reducible) words from $R_i$ as words from a context-free grammar and use the methods of https://linkinghub.elsevier.com/retrieve/pii/0020019094900337 to sample uniformly from it with a better control over the length.
>
> **this notion of "variability" has not been defined**
>
> To provide a more precise definition, we can define variability as the ratio of the number of unique words generated per batch to the total number of words of a given length. This metric is detailed in (new) Table 1.
>
> **why the distribution (of valleys) for the bracket-style sampling is "better" than naive**
>
> The distribution for bracket style is better because it is "closer to uniform". But this is necessary condition for the ideal sampler, not sufficient. Perhaps the better illustration to illustrate the benefits of bracket style sequencing would be a plot like Figure 1 in the attachment. In general, in the paper the number of valleys serves just as an auxilary statistic of the sampler and the whole discussion of Dyck paths and Dyck languages serves an additional purpose to tie our research with the research on generating Dyck languages, as stated in the paper.
>
> **Could you provide quantitative evidence that the LLM results are "covering" the space well?**
>
> We could not provide such evidence now. For us the "degenerate pattern" you mention is defined as being an element of $[R_0\dots R_n]_S$, hence this question is more suitable for us in a context of negative sampling, where we train the model to generate elements away from $[R_0\dots R_n]_S$. Note that both $\cap_iR_i$ and $[R_0\dots R_n]_S$ are infinitely generated subgroups of free group, but their quotient (i.e. the homotopy group itself) is finite (provided $n > 2$). To our knowledge there is no known purely group-theoretical proof of this fact.
>
>
> **... if we limit ourselves to words with length less than some threshold?**
>
> Because of the remark above, if we are studying the question of coverage in a context of generating from intersection (not the homotopy group), we should restrict ourself to some finite subset of the free group. The word length restriction may not be natural for the type of objects like $\cap_iR_i$, in a sense that it is quite tricky to get a formula for number of words in $R_i$ of a given length, not to mention $\cap_iR_i$. We can bruteforce such computation for one $R_i$ and smaller lengths (used in Table 1 in the attachment), but these are insufficient to draw conclusions about the full intersection. We are investigating other filtrations on the free group that may be more natural for the problem.
>
> **there is potential for train/test overlap?**
>
> As mentioned above, we check that the words in the training datasets are not from the full intersection, hence train/test overlap is not possible. Additionally we can ensure the non-overlapping of the train and the test by separating a subset of prefixes, that would only be shown to the model during the test phase.

---

> > ### Comment · Reviewer_LHhA · 2023-11-22
> > **Thank you for your reply, but my concerns remain**
> >
> > Thank you for your reply.
> >
> > Your comments on the reasons for bracket over naive, as well as the associated supplementary PDF, made things quite a bit clearer on that front.
> >
> > Unfortunately, in my opinion, this work does not demonstrate enough significance in either a machine learning or mathematical setting. Your reply states that your goal was to transition the problem of computing homotopy groups into a domain of machine learning, however I do not see your current work providing a practical solution here, for the reasons I originally outlined in the section on weaknesses. In particular, the work does not generate elements from the homotopy group but rather just from the full intersection of normal closures, and it is unclear on how this procedure provides any insight into the structure of the homotopy group. In addition, there is the lack of any notion of "coverage", and so even in the context of generating elements from the full intersection it is not clear what conclusions could be drawn from such a process.
> >
> > Overall, my impression is that this work is a first attempt at the following idea: use generative models and prompt strategies to generate elements of free groups which are otherwise difficult to sample from. This idea is certainly novel, however for the reasons above it seems a bit too limited to be of practical use for the proposed task.

---

### Official Review · Reviewer_aKQP · 2023-11-01

**Soundness:** 4 excellent
**Presentation:** 4 excellent
**Contribution:** 4 excellent
**Rating:** 6
**Confidence:** 3

**Summary:**

This paper aims at studying the simplicial homotopy groups of the 2-dimensional sphere. Following Wu's formula, each homotopy group is isomorphic to the quotient of a free group. Each element in the homotopy group, as a reduced word, can be connected to a sentence in the language models. The decoder part of a transformer is used to generate approximate samples from the homotopy group, and a multi-label is used to denote whether the sample satisfies relations of the homotopy groups. Multiple approaches for handling the multi-label and training the transformer are designed. Adequate numerical experiments are provided, with comparison to multiple baselines.

**Strengths:**

1. The problem originating from algebraic topology is important.
2. The idea is very innovative.
3. The paper is well-written and mostly clear.
4. Adequate numerical experiments are provided.

**Weaknesses:**

1. It would be helpful to discuss a bit of how the samples drawn from the homotopy group can be used to understand the topological properties of the 2-dimensional sphere, compared to the more obtainable homology groups.
2. It would be nice to discuss the significance and potential applications of the proposed method to topological spaces other than the 2-dimensional sphere.

**Questions:**

1. Does the samples drawn using the decoder transformer have a much wider diversity compared to the baseline methods? What are the proportions of repeated samples?
2. Is it possible to intuitively understand the distributions over the homotopy group that the transformer or the baseline methods are drawing from?

---

> ### Author Response · Authors · 2023-11-20
>
> **The decoder part of a transformer is used to generate approximate samples from the homotopy group**
>
> It is important to clarify that in this project, our focus has been exclusively on generating samples from the numerator of the Wu formula. The generation of non-trivial elements from the homotopy group itself is a significantly more challenging problem. Our current research is laying the groundwork for this, and we are actively working on it in our current project.
>
>
> **how the samples drawn from the homotopy group can be used to understand the topological properties**
>
> Almost nothing is known about group-theoretic description of elements of the homotopy groups in terms of their topological properties. We can speculate that the commutator $[x_1, x_2]$ which generates $\pi_3 S^2$ and corresponds to the Hopf fibration $S^1 \to S^3 \to S^2$ is related to the Whitehead product expression for it. Deeper connections can be given using the language of $\Lambda$-algebra as well.
>
>
> **spaces other than the 2-dimensional sphere**
>
> Our results can be applied to spaces which have infinite cellular decompositions, but still fit into the framework of Wu's formula. Such spaces are usually unreachable by methods of the classical computational topology. Example os such space is $\Sigma\mathbb R \mathrm{P}^\infty$, i.e. a suspension over the infinite real projective plane. We are investigating this particular example currently.
>
>
> **proportions of repeated samples?**
>
> Our metrics count the numbers of unique generated words. We will count the repetitions as well and report them in the supplementary material if time permits.
>
>
> **distributions over the homotopy group ?**
>
> As for now we do not know how to describe this distribution. This problem is quite challenging since the answer is unknown theoretically (in the group-theoretic language) and the full intersection that we are sampling from is infinitely generated (see also answer to LHhA).

---

> > ### Comment · Reviewer_aKQP · 2023-11-20
> >
> > Thank you to the authors for the detailed reply and clarification. I myself have enjoyed the idea behind this paper although may not fully grasp certain intricate math details. Contrary to the other reviewer's comments, I find this paper to be quite motivating and is one of the good examples of using machine learning ideas in studing algebraic topology, and should qualify to be at a top machine learning conference. I am keeping my rating as is.

---

### Author Response · Authors · 2023-11-20

We would like to thank our referees for their time and effort in reviewing our paper. Thank you very much for highly value the originality of our work! We apologize for our oversights on the presentation and writing (like forgetting to define the subgroup $[R_i, R_j]$), we will fix these at once.


We have updated the supplementary material and included the file $\verb|iclr_2024_rebuttal.pdf|$, which contains Figure 1 and Table 1 with the results of the experiments clarifying some questions on bracket-style vs. naive sampling. If time permits, we will include additional experiments as well.

---

### Meta-Review · Area_Chair_CtLV · 2023-12-06

**Metareview:**

This paper presents a novel approach of using language modeling techniques to tackle the challenge of sampling from the simplicial homotopy groups of topological spaces. The technique harnesses the decoder part of a transformer to generate approximations of elements from the homotopy group of the 2-dimensional sphere, positioning machine learning as a bridge to comprehend elements of algebraic topology.

The paper's strengths lie in its originality and the successful application of established ML architectures to a new domain. Adequate numerical experiments substantiating the claims have been provided and are consistently better than compared baselines.

However, the reviewers have highlighted several concerns that must be addressed. The reviewer LHhA emphasized the unclear applicability of the current framework, particularly the work does not generate elements from the homotopy group but rather just from the full intersection of normal closures, and it is unclear on how this procedure provides any insight into the structure of the homotopy group. Additionally, the manuscript would benefit from a more thorough discussion on how these generated samples can shed light on the topological properties of spaces.

The presentation was also a point of concern among several reviewers, who have noted that the paper might be too specialized for the broader machine learning audience. The lack of clarity in the application of ML methods and the highly technical language pose significant barriers for understanding and suggests that there is a need for further refinement to accommodate interdisciplinary discourse.

In conclusion, the paper offers a promising interdisciplinary application of machine learning to a complex field. However, its contributions would be more impactful with significant improvements in the presentation to appeal to the ML community, explicit implications or discussions of the contributions from a mathematical perspective, and a clearer demonstration of its practical value. Considering these aspects, as the meta-reviewer, I am in agreement with the reviewers that the paper is not ready for publication at ICLR in its current form. The authors are encouraged to take the reviewers' feedback into consideration for future submissions or journal publications where the work might find a more specialized audience.

**Justification For Why Not Higher Score:**

2 of 3 reviewers have severe concerns that the significance of this work is unclear and  the work needs to be expanded in order to be appropriate for ICLR.

**Justification For Why Not Lower Score:**

N/A

---

### Decision · Program_Chairs · 2024-01-16

Reject